# Design and Analysis of the Cis-Lunar Navigation for the ArgoMoon CubeSat Mission

Marco Lombardo [1,*], Marco Zannoni [1,2], Igor Gai [1], Luis Gomez Casajus [2], Edoardo Gramigna [1], Riccardo Lasagni Manghi [1], Paolo Tortora [1,2], Valerio Di Tana [3], Biagio Cotugno [3], Simone Simonetti [3], Silvio Patruno [3] and Simone Pirrotta [4]

1 Dipartimento di Ingegneria Industriale, Alma Mater Studiorum—Università di Bologna, Via Fontanelle 40, 47121 Forlì, Italy
2 Centro Interdipartimentale di Ricerca Industriale Aerospaziale (CIRI AERO), Alma Mater Studiorum—Università di Bologna, Via Baldassarre Carnaccini 12, 47121 Forlì, Italy
3 Argotec S.r.l., Via Cervino 52, 10155 Torino, Italy
4 Agenzia Spaziale Italiana, Via del Politecnico, 00133 Roma, Italy
* Correspondence: marco.lombardo14@unibo.it

**Abstract:** In the framework of the Artemis-1 mission, 10 CubeSats will be released, including the 6U CubeSat ArgoMoon, built by the Italian company Argotec and coordinated by the Italian Space Agency. The primary goal of ArgoMoon is to capture images of the Interim Cryogenic Propulsion Stage. Then, ArgoMoon will be placed into a highly elliptical orbit around the Earth with several encounters with the Moon. In this phase, the navigation process will require a precise Orbit Determination (OD) and a Flight Path Control (FPC) to satisfy the navigation requirements. The OD will estimate the spacecraft trajectory using ground-based radiometric observables. The FPC is based on an optimal control strategy designed to reduce the dispersion with respect to the reference trajectory and minimize the total ΔV. A linear approach was used to determine the optimal targets and the number and location of the orbital maneuvers. A covariance analysis was performed to assess the expected OD performance and its robustness. The analysis results show that the reference translunar trajectory can be successfully flown and the navigation performance is strongly dependent on the uncertainties of the ArgoMoon's Propulsion Subsystem and of the orbit injection.

**Keywords:** orbit determination; CubeSat; ArgoMoon; deep space navigation; flight path control; artemis

## 1. Introduction

The CubeSat standard originated as a cost-effective alternative for relatively simple and short missions in low Earth orbit [1]. Over the years, these platforms have been extensively tested in space and have proved a reliable technology capable of performing many tasks typical of a "traditional" space mission. Based on these advances, the idea of using one or more CubeSats as companions to a traditional deep space probe was proposed. For instance, small satellites could be used for secondary mission tasks, to perform gravity investigation using formation flight (Hera-Juventas [2]), or dedicated optical observation (LICIACube [3], ArgoMoon [4]). However, in the current state of the art, small satellites have only been tested twice in deep space (NASA's MarCO [5] and ASI's LICIACube). Therefore, extensive studies, analyses, and experimental tests must be conducted to successfully operate small satellites beyond the Low Earth Orbit (LEO) [6]. Moreover, the exploration of the Moon attracted relevant international interest in recent years. This interest led to the proposal and selection of many small satellite missions that are currently flying or will be launched in the next future (e.g., CAPSTONE [7], LUMIO [8], HORYU-VI [9]).

The ArgoMoon mission will participate in this innovation context as the first European CubeSat that will fly in a cis-lunar orbit as a secondary payload of the newborn NASA's

SLS on its maiden flight. The mission aims to autonomously fly the ArgoMoon CubeSat around the Interim Cryogenic Propulsion Stage (ICPS) to capture detailed and significant pictures of the stage and confirm the deployment of the other CubeSats during the first 6 h of the mission [4]. Then, the mission foresees flying ArgoMoon for 180 days in a highly elliptical orbit around the Earth. In this phase, the CubeSat platform and the ground operations, including navigation, will be intensively tested as a technological demonstration. The navigation in deep space of CubeSats poses additional challenges with respect to a traditional mission, due to onboard resource limits, miniaturized thrusters, and smaller non-directional antennas. This manuscript describes the design and performance characterization of the navigation of the ArgoMoon mission, and it is organized as follows: Section 2 describes the ArgoMoon mission, focusing on the navigation constraints and requirements; the Flight Path Control (FPC) analysis, designed to control the trajectory, is outlined in Section 3, while the Orbit Determination (OD), needed to estimate and predict the orbital evolution, is covered in Section 4. Section 5 presents a sensitivity analysis to assess the robustness of the designed navigation process. Finally, Section 6 summarizes the main findings and conclusions of this work.

## 2. The ArgoMoon Mission

### 2.1. Mission Overview

The Artemis-1 mission foresees the first launch of the heavy-lift rocket SLS that will inject the Orion Multi-Purpose Crew Vehicle (MCPV) in its first flight to the cis-lunar region [10]. At present, the Artemis-1 launch date is set to be Not Earlier Than (NET) 12 November 2022. As secondary payloads of the SLS, a total of 10 CubeSats have been integrated into the ICPS by NASA to carry out scientific experiments and perform technology demonstrations [11]. The CubeSats are scheduled for deployment from the ICPS at five specific trajectory locations named Bus Stops (BS) [12]. The deployment phase will begin once MCPV separates from the ICPS, after the insertion in the cis-lunar trajectory.

ArgoMoon is one of the first CubeSats to be released at the BS-1 (first deployment stage) which will occur approximately 3 h and 54 min after the launch [4]. The ArgoMoon mission will start once the spacecraft (S/C) is released from the ICPS dispenser, and the deployment epoch $T_D$ is considered the beginning of the timeline of the operations. The mission is subdivided into three main phases of different durations. Phase 1, which covers the first day of flight, is the most critical since ArgoMoon will perform the automatic proximity flight operations (ProxOps) close to the ICPS. The S/C will operate autonomously for 30 min after the deployment without s direct ground link. The relative positioning and pointing with respect to ICPS is performed by ArgoMoon using pictures acquired on board and processed by an image recognition algorithm based on machine learning [4]. Then, the S/C will turn on its onboard radio and the link with the antennas of the DSN will be established, allowing to perform ground-based orbit determination and navigation through radiometric observables [13]. After Phase 1 is completed, an orbital maneuver will be executed to target the first fly-by of the Moon and shape the geocentric trajectory. Phase 2 covers up to 20 days after the S/C deployment, and it starts once the first orbital maneuver is completed. During Phase 2, ArgoMoon will then perform the first fly-by of the Moon and its first revolution around the Earth. Phase 3 starts 20 days after the deployment, and it covers the remainder of the mission. During the last week of Phase 3, about 180 days after the deployment, ArgoMoon will perform a final fly-by with the Moon to be injected into a heliocentric orbit for its final disposal. The End of Mission (EOM) will occur when the S/C reaches the heliocentric orbit and the tracking activities will be terminated.

The S/C will be managed and operated from Turin (Italy) at the Argotec Mission Control Center (MCC), where the navigation will be performed by personnel from the Radio Science and Planetary Exploration Laboratory of the University of Bologna (Website link: www.site.unibo.it/radioscience-and-planetary-exploration-lab, accessed on 20 October 2022).

### 2.2. The Spacecraft

ArgoMoon is a 6U CubeSat based on the HAWK-6 platform, Figure 1, developed and assembled by Argotec [4]. The platform maximum mass is 14 kg, and it is powered by a Solar Panel Array (SPA) designed as a couple of double-side retractable wings that provides up to 80 W. The orbital propulsion is provided by one main monopropellant thruster with 0.1 N of nominal thrust and 192 s of specific impulse. The maximum expected ΔV that the monopropellant thruster will deliver to the S/C is 57 m/s. The attitude is controlled with a 3-axis stabilization system equipped with a star tracker, sun sensors, an Inertial Measurement Unit (IMU), and three reaction wheels.

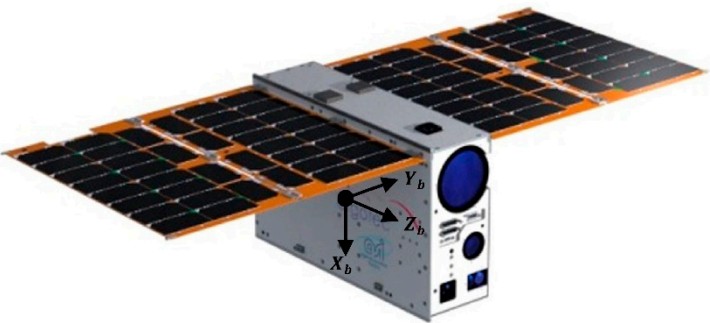

**Figure 1.** ArgoMoon CubeSat external view render adapted with permission from Ref. [4]. Copyright year, IEEE and the assumed body-fixed frame.

The attitude control during the execution of an orbital maneuver and the reaction wheels desaturations are carried out by a Reaction Control System (RCS) based on four double-canted cold-gas thrusters with 0.057 N of nominal thrust. The Telemetry Tracking & Command (TT&C) subsystem is based on the miniaturized X-Band (7.2–8.4 GHz) IRIS transponder of NASA/JPL [14]. The IRIS is connected to four patch antennas on two opposite sides of the spacecraft (−X and +X faces). The ArgoMoon's payload includes one narrow- and one wide-angle optical camera and a laser Rangefinder capable of tracking a target up to 5 km of distance [4]. The narrow-angle camera is named Payload 1 (PL1), and it has a Field of View (FOV) of 2.05 deg. The wide-angle camera has a FOV of 32.5 deg and it is identified as PL2. The payloads are the core sensors used by the onboard Image recognition Software (IS) to recognize, track, and point different objects (i.e., the ICPS, Earth or Moon) lying in the Field of View of the cameras [4].

### 2.3. Trajectory

The ArgoMoon's reference trajectory is designed by Argotec with the aim of flying the S/C between the cis-lunar and the translunar space for 180 days. The shift of the SLS launch dates had only a negligible impact on the trajectory geometry as well as on the operations timeline. The analysis discussed in this paper, based on the launch window of 6 June 2022, can be considered for past and future launch dates. The reference trajectory has been designed as a highly elliptical geocentric orbit characterized by two close fly-bys of the Moon, one at the beginning of the mission and the other at the end. The first fly-by (M0) occurs approximately five days after the deployment, and it is used by ArgoMoon to shape its orbit around the Earth. The other fly-by of the Moon (M3) is designed to inject the S/C into a heliocentric orbit for its disposal at EOM. In between, there are also two secondary fly-bys of the Moon (M1 and M2), with a Closest Approach (C/A) at a very high altitude (above 80,000 km), which makes them not critical for the mission. The trajectory is displayed in Figure 2, and the fundamental mission events are summarized in Table 1.

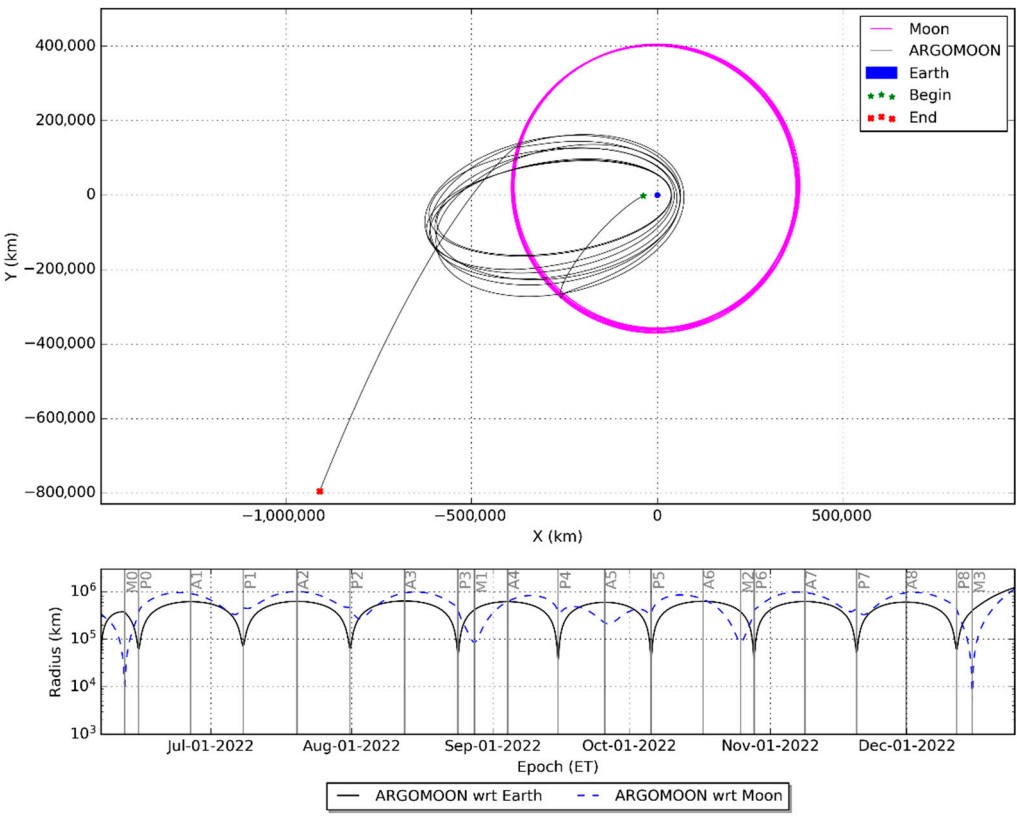

**Figure 2.** ArgoMoon trajectory in the Earth Mean Orbit frame (at J2000 epoch) and the S/C distance with respect to the Earth and the Moon.

The trajectory begins at $T_D$, where the real initial state vector after the deployment cannot be fully characterized *a priori* because the ICPS will rotate around its primary inertia axis before initiating the dispensing activities. However, since the uncertainty introduced by the deployment is negligible if compared to the ICPS injection covariance, the S/C initial state has been assumed to be equal to the one of ICPS at $T_D$. After the deployment, ArgoMoon will follow its release path for 15 s and then it will perform a rotation of 180 degrees to point the PL2 towards ICPS for the autonomous target tracking. Then, during the first 30 min of flight, the S/C will perform two orbital maneuvers to maintain a close and stable distance to the ICPS. However, the uncertainties introduced by the automatic orbital maneuvers are negligible with respect to the ICPS injection covariance. Therefore, the ProxOps phase around the ICPS during the first 30 min has not been modeled. Seventy-five minutes after the deployment, ArgoMoon will perform the Keep Out Zone (KOZ) maneuver required to drift away from the ICPS and then start Phase 2 of the mission. The KOZ is a one-second impulse executed aligned with the ArgoMoon-ICPS line-of-sight but in the opposite direction with respect to the ICPS. At 20 h after the deployment, an Orbit Trim Maneuver (OTM), named OTM1, with a ΔV of 10.95 m/s, is executed to target the fly-by M0 and shape the later geocentric trajectory. Nominally, no other deterministic orbital maneuvers are planned for ArgoMoon's reference orbit. The trajectory foresees eight revolutions (REVs) around the Earth where the perigees and apogees are identified using the capital letters P and A followed by an incremental number starting from zero (i.e., P0 is the first perigee, A8 is the last apogee). A REV is defined between two successive perigees (i.e., REVi is between Pi and Pi + 1), except for REV0 and REV9 at the beginning and at the end of the mission, where M0 and M3 take place and there is no initial or final perigee. Before M3, the maximum distance from the Earth that ArgoMoon will reach in its path is 830,000 km, where the closest one is 37,400 km.

**Table 1.** ArgoMoon summary mission timeline for the launch date of 6 June 2022.

| Event | Event Epoch | Details |
|---|---|---|
| Bus Stop 1 (BS1) | Launch + 3 h 54 min | First CubeSats dispensing phase |
| Bus Stop 2 (BS2) | Launch + 6 h 59 min | Last ArgoMoon observed deployment phase |
| Deployment ($T_D$) | BS1 + 6 min | Release of ArgoMoon from the ICPS (close to BS1) |
| Transponder ON | $T_D$ + 30 min | ArgoMoon starts to communicate with DSN |
| KOZ | $T_D$ + 75 min | Keep Out Zone maneuver to drift away from the ICPS |
| OTM1 | ~$T_D$ + 20 h | Maneuver to trim the first fly-by of the Moon (M0) |
| M0 | ~$T_D$ + 5.23 days | First fly-by of the Moon: C/A at 7773 km |
| M1 | ~$T_D$ + 82.08 days | Mid-course fly-by of the Moon: C/A at 86,051 km |
| M2 | ~$T_D$ + 104.61 days | Mid-course fly-by of the Moon: C/A at 84,594 km |
| M3 | ~$T_D$ + 191.51 days | Last fly-by of the Moon: C/A at 5261 km |
| EOM | $T_D$+200 days | End of the mission |
| Pi (i = 0,1 … 8) | Perigees | Total number of perigees: 9 |
| Ai (i = 0,1 … 8) | Apogees | Total number of apogees: 9 |
| REV0 | $T_D$ to P0 | First revolution that encompasses the fly-by M0 |
| REVi (i = 1,8) | Pi to Pi + 1 | Revolutions around the Earth (i.e., REV3: from P2 to P3) |
| REV9 | P8 to EOM | Last revolution that encompasses the fly-by M3 |

*2.4. Navigation Requirements*

The primary objectives of the mission, as seen in Section 2.1, are to fly ArgoMoon safely around the ICPS to acquire significant photographic documentation of the upper stage, extensively test the CubeSat platform and dispose the S/C in heliocentric orbit at the EOM. The ProxOps and image acquisition around the ICPS will be mainly guaranteed by the ArgoMoon autonomous flight control algorithm, without directly involving the navigation team. Moreover, no other strong requirements have been placed for the remainder of the mission. Therefore, in the analysis proposed in this paper, a set of requirements was identified to guarantee the navigability of the designed reference trajectory:

- **Impact avoidance**: the S/C shall not fly below the threshold altitudes of 1000 km with respect to the Earth and 100 km with respect to the Moon. The requirement applies to the whole mission and can become significant at the perigees and fly-bys of the Moon.
- **Heliocentric disposal**: the S/C shall reach the heliocentric disposal orbit after the last fly-by of the Moon. The ranges of tolerance for the disposal conditions have been determined through a Monte Carlo analysis with the requirement of having a low probability of crossing the Earth's sphere of influence in successive years. The disposal requirement is displayed in Figure 3, where the green dots are the samples with a correct disposal and the red crosses are the ones that do not satisfy the requirement.
- **DSN pointing uncertainty**: to ensure the link with the DSN 34 m antennas, the pointing uncertainty due to S/C orbit determination shall be lower than 0.031 deg, which corresponds to the Half Power Beamwidth (HPB) of the antenna at X-band [15]. However, during the first day of the mission, the threshold value of the pointing uncertainty is relaxed to 1.05 deg, which corresponds to half of the HPB of the 34 m dishes equipped with the 1.2 m aided acquisition antenna above the sub-reflector [15].

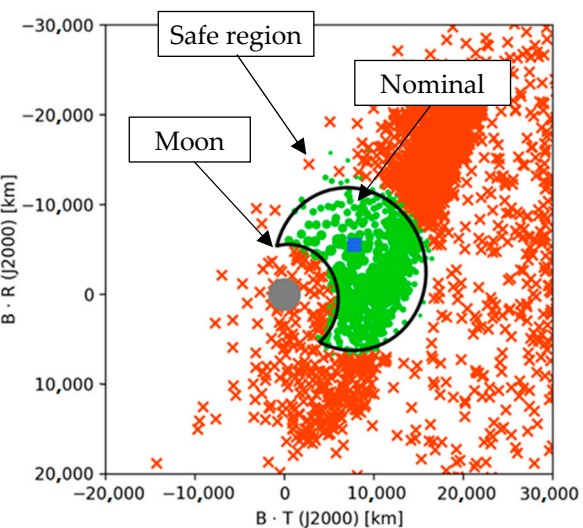

**Figure 3.** B-Plane [16] admissible region for the last fly-by of the Moon (M3).

All requirements have to be satisfied with a 3-sigma confidence, or higher than 99% if the distribution is not Gaussian. After Phase 1, the available ΔV allocated to the navigation is expected to be 57 m/s.

*2.5. Navigation Concept*

ArgoMoon's navigation process is divided into Orbit Determination (OD) and Flight Path Control (FPC). The OD provides an estimation of the spacecraft state vector and other dynamical model parameters (such as orbital maneuvers), by iteratively correcting an *a priori* dynamical model to minimize the so-called residuals using a batched least-square filter [17]. The residuals are the differences between the real observed quantities, and the ones computed using the *a priori* dynamic model. The main observables that will be used in the OD of ArgoMoon are two-way coherent Doppler and two-way range, and one-way Doppler [18]. The Doppler observable provides a high-resolution and accurate measurement of the line-of-sight velocity of the S/C with respect to the antenna on the ground. The Doppler accuracy level achievable using the IRIS transponder by tracking from the DSN antennas is expected to be in the order of 0.1 mm/s, at 60 sec integration time [13,19]. The range observable provides the line-of-sight distance between the S/C and the ground antenna, with a typical accuracy of a few meters. The radiometric data will be corrected for the path delay induced by Earth's troposphere and ionosphere using standard GNSS-based calibrations provided by the DSN [20]. The OD process is nominally performed after each Data Cut Off (DCO), which is identified as the epoch of the last measurement of the dataset used in the analysis. The OD products, which contain the best estimate of the S/C trajectory, can be used in the FPC process to compute the trajectory correction maneuvers required to follow the planned path. In this case, the DCO used to generate the OD products is named "maneuver's DCO". The primary sources of errors affecting the S/C trajectory are the launcher performance, the S/C thrusters' performance, the dynamical mis-modeling, and the OD errors. Hence, due to the non-deterministic nature of the trajectory errors, the correction maneuvers are purely statistical and cannot be computed *a priori* before the beginning of the mission. Thus, in the ArgoMoon navigation process, the orbital maneuvers are distinguished in OTM and Statistical Trim Maneuver (STM). An OTM is a deterministic burn that has been computed *a priori* by the mission analysis in the design of the reference trajectory. An STM is an orbital maneuver scheduled *a priori* through the navigation analysis, whose real ΔV can only be computed during the operations. The orbital maneuvers computed by the FPC are then processed by the MCC operators and converted into telecommands for the S/C. Finally, the trajectory estimated

by the OD is used to update the onboard S/C ephemeris and used by the DSN to point the antennas.

ArgoMoon's navigation will be performed by dividing the trajectory into single arcs, encompassing a single REV, to be processed sequentially. The navigation system and the pre-launch analysis have been designed and performed using NASA/JPL's navigation software, the Mission Analysis, Operations, and Navigation Toolkit Environment (MONTE) [21]. The software MONTE is currently (or has been) used for the operations in NASA's deep-space missions, and for radio science experiments data analysis [22–25].

## 3. Flight Path Control Analysis

The assessment of the navigation feasibility requires evaluating the amount of propellant needed to correct the errors that may potentially affect the trajectory. An optimal trajectory control strategy is then required to correct the statistical errors without exceeding the $\Delta V$ allocated to the navigation. The FPC analysis of this paper was performed to design a control strategy that significantly reduces the dispersion with respect to the reference trajectory while minimizing the number of maneuvers and the total required $\Delta V$. The study was conducted using a linear analysis to test different maneuver schedules, aimpoints, and target coordinates [26]. Then, once an optimal solution was found, a non-linear analysis was carried out to validate the trajectory dispersion and the statistical $\Delta V$. The linear method uses the covariance mapped to future times to evaluate the dispersion and the controllability of the selected targets. The non-linear approach is based on the numerical propagation of the trajectory using the same high-fidelity model described in Section 4.2. Both simulations were performed using the Monte Carlo method, which samples from the covariances designed to describe the expected sources of errors. The considered sources of error and the relative assumptions are summarized in Table 2. The wide uncertainty on the SLS performance and ArgoMoon propulsion system capabilities are expected to be the predominant sources of error.

**Table 2.** Considered trajectory statistical errors in the Flight Path Control analysis.

| | ICPS state (Earth-RTN) uncertainty (3-sigma) at BS1 epoch: | | | | | |
|---|---|---|---|---|---|---|
| Injection covariance | X (km) | Y (km) | Z (km) | VX (km/s) | VY (km/s) | VZ (km/s) |
| | 30.0 | 60.0 | 15.0 | 0.0021 | 0.0027 | 0.0042 |
| Maneuvers execution error | Gates Model applied to both OTMs and STMs. | | | | | |
| Mis-modeling and OD error | OD covariance mapped from the maneuver's DCO to the aimpoint. | | | | | |

The dispersion on the S/C initial state is performed by sampling the position and velocity errors from the ICPS injection covariance. The maneuver's execution error is implemented using the Gates model [27]. The model is used to generate Gaussian randomly sampled execution errors for a specific commanded $\Delta V$, both in magnitude and in pointing. The generated errors are then added to the commanded maneuver to simulate the real maneuver. Table 3 summarizes the Gates model parameters assumed in this work. The OD error is simulated using the OD covariance mapped from the maneuver's DCO to the target's epoch to generate a randomly sampled error in the maneuver's target values. The mis-modeling is included in the OD error computation by estimating stochastic accelerations during the OD simulation.

**Table 3.** Gates model assumptions for the maneuvers execution error (1σ).

|  |  | Error Component (Per Axis) | ArgoMoon PS |
|---|---|---|---|
| Maneuvers execution error | Magnitude | Fixed (m/s) | 0.011 |
|  |  | Proportional (%) | 3.5 |
|  | Pointing | Fixed (m/s) | 0.011 |
|  |  | Proportional (deg) | 1.1 |

### 3.1. Uncontrolled Trajectory

At first, a Monte Carlo analysis of the uncontrolled trajectory has been performed to evaluate the sensitivity of the trajectory to the initial conditions. Moreover, the analysis allowed to evaluate the Earth and Moon impact risks and the heliocentric disposal probability without performing any correction maneuver. A total of 10,000 trajectory samples were propagated starting from the reference initial state, perturbed using the injection covariance.

Then, the OTM1 is executed by adding the maneuver execution error and the trajectory is propagated up to the EOM without simulating any other correction maneuvers. For each trajectory sample, the position and velocity error with respect to the reference trajectory is computed to generate the statistics of the dispersion. The results in Figure 4 show the magnitude of the position error computed with respect to the nominal trajectory up to A1. Figure 5 reports the results of the trajectory dispersion mapped on the B-Plane [16] of the fly-by M0. The plots have been limited only to the first relevant part of the mission due to the chaotic behavior of the solutions after P0. The initial error caused by the injection covariance and the OTM1 execution error quickly expands after the closest approaches with the Moon (M0) and the Earth (P0). Then, it remains bounded between $10^5$ km and $10^6$ km up to the end of the mission. The uncontrolled dispersion causes a probability of about 1.8% to fly below 1000 km of altitude with respect to the Earth, violating the impact risk requirement. Moreover, due to the chaotic behavior of the propagated samples, the last fly-by with the Moon (M3) is never achieved.

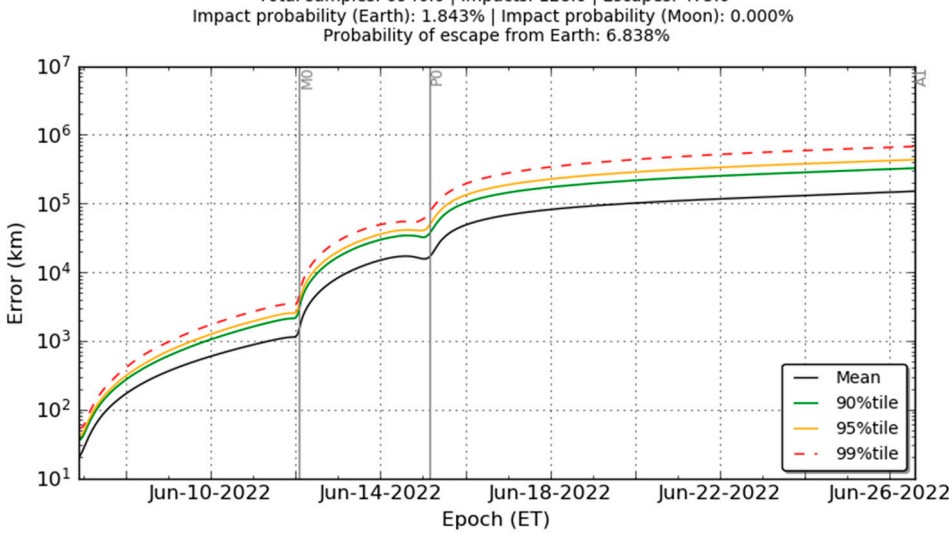

**Figure 4.** Statistics of the uncontrolled trajectory error with respect to the nominal trajectory from the deployment up to the first apogee A1.

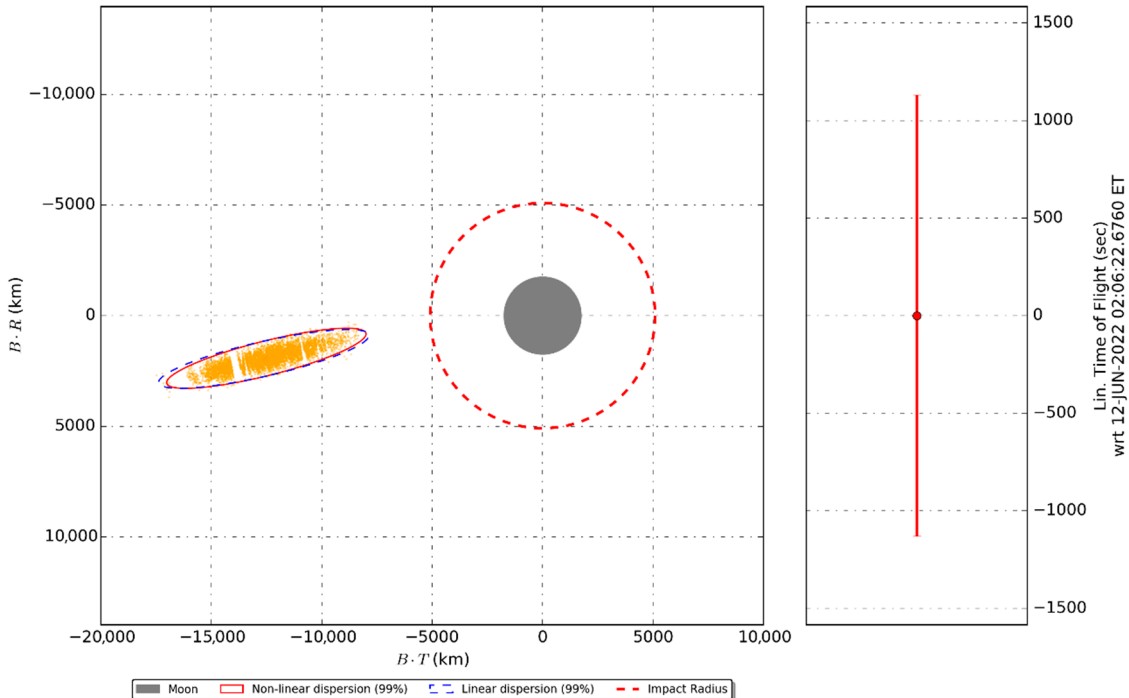

**Figure 5.** The 99th percentile of the trajectory dispersion on the B-Plane [16] of M0.

The large dispersion after M0 is an indicator that the first fly-by of the mission is critical, and the trajectory is very sensitive to the initial conditions, as expected for three-body problems. In fact, due to the shape of the orbit, the third body perturbations of the Sun and the Moon are predominant and, as it is well known for the three-body problems, even a small perturbation on the initial conditions may lead to a very different solution.

### 3.2. Optimal Control Strategy

To satisfy the navigation requirements, the dispersion with respect to the reference trajectory must be minimized. This required searching for a set of aimpoints to target through impulsive orbital maneuvers. The selection of the aimpoints and the targeting maneuver locations were evaluated using the K-inverse method [26]. The K matrix is defined as a $3 \times 3$ block of the State Transition Matrix (STM), mapped to the aimpoint's epoch in the desired coordinate system, extracted as

$$K = \begin{bmatrix} \frac{\partial c_1(t_A)}{\partial \dot{x}(t)} & \frac{\partial c_1(t_A)}{\partial \dot{y}(t)} & \frac{\partial c_1(t_A)}{\partial \dot{z}(t)} \\ \frac{\partial c_2(t_A)}{\partial \dot{x}(t)} & \frac{\partial c_2(t_A)}{\partial \dot{y}(t)} & \frac{\partial c_2(t_A)}{\partial \dot{z}(t)} \\ \frac{\partial c_3(t_A)}{\partial \dot{x}(t)} & \frac{\partial c_3(t_A)}{\partial \dot{y}(t)} & \frac{\partial c_3(t_A)}{\partial \dot{z}(t)} \end{bmatrix}$$

where $c_1$, $c_2$ and $c_3$ are three targeted coordinates at the aimpoint epoch $t_A$ and $\dot{x}(t)$, $\dot{y}(t)$ and $\dot{z}(t)$ are the S/C velocity components at an epoch $t < t_A$. The K matrix allows to evaluate the sensitivity of the targeted coordinates with respect to a change in the orbital velocity. Therefore, if $\Delta X(t_A)$ represents a perturbation of the aimpoint coordinates $c_1$, $c_2$ and $c_3$, the variation in the S/C velocity required to correct this perturbation can be computed from the inverse of K as

$$\Delta V(t) = K^{-1} \Delta X(t_A)$$

Then, searching for the minimum of the K-inverse norm provides the optimal maneuver, which minimizes the $\Delta V$. The use of a linear method has given the possibility to rapidly test many kinds of aimpoints and coordinates to be targeted, as well as the maneuver's

location. Based on previous studies, the optimal aimpoints were selected to be the fly-bys with the Moon (M0, M3) and apocenters (A1, A2, . . . , A8) of the orbit around the Earth [26]. Moreover, to reduce the trajectory dispersion by improving the control on the S/C orbital velocity, the pericenters (P1, P2, . . . , P7) of the orbit around the Earth have been selected as further aimpoints.

The fly-bys with the Moon are targeted using the B-Plane coordinates B.R, B.T and the Linearized Time of Flight (LTOF) [16]. The targeted coordinates of the apsides aimpoints are the cartesian position at the apogees and the cartesian velocity at the perigees.

The K-inverse norm evolution for the fly-bys M0 and M3 are shown in Figure 6. Before M0, the minimum of the norm is located approximately where the OTM1 has been placed by the mission analysis team. Then, moving toward M0, the norm becomes larger, causing any maneuvers close to the fly-by to be more expensive in terms of ΔV. For this reason, STM1 was placed 48 h after OTM1, as early as possible, but still keeping a sufficient margin to perform the navigation process. The norm analysis of M3 was performed starting from the beginning of the REV8. The results highlight two local minima of the norm, approximately one at two days after the apogee A8 and the other 12 h after the last perigee P8, where the correction maneuvers STM18 and STM19 should be placed.

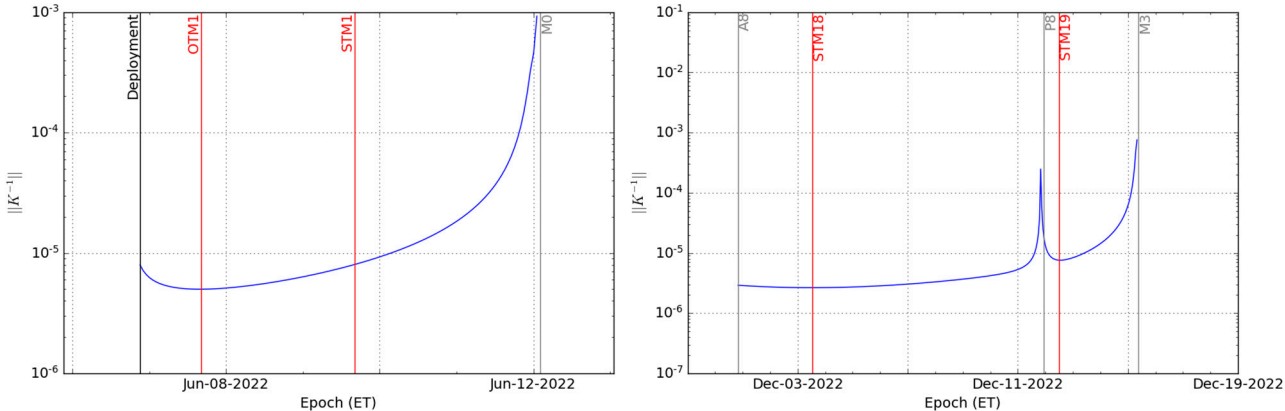

**Figure 6.** K-inverse norm evolution for the targeting of B.R, B.T and LTOF coordinates of the B-Plane for the fly-bys M0 (**left**) and M3 (**right**).

Figure 7 depicts the evolution of the K-inverse norm during REV1, aiming at the cartesian position at A1 and cartesian velocity at P1. The targeting of the cartesian position at A1 is optimal if performed at the local minimum two days after the perigee P0. Close to perigees and apogees, there is a local maximum due to the 180-degree transfer singularity of the Lambert problem [26]. The norm evolution for the cartesian velocity at P1 shows a minimum close to A1, even if the curve is relatively flat. Close to the perigees, the norm increases because of the reduction of the orbital velocity. The same considerations apply to the orbits REV2 to REV8.

Finally, using the found optimal locations and targets for the STMs, a Monte Carlo non-linear analysis has been performed to assess the expected statistical ΔV and the controlled trajectory dispersion. The statistical ΔV is reported in Table 4 in terms of mean, i.e., the expected value, and the 99th percentile, used for the requirements validation. The ΔV results show that STM1, STM2, STM3, and STM4 are the most expensive statistical maneuvers. Observing the controlled dispersion in Figure 8, the peaks on the position errors with respect to the reference trajectory occur during REV0, especially after M0, reaching a maximum value at P0 of about 3000 km. After A1 and before M3, the dispersion always remains below about 400 km (at perigees), increasing again to 1000 km only at EOM because of the disposal flyby M3.

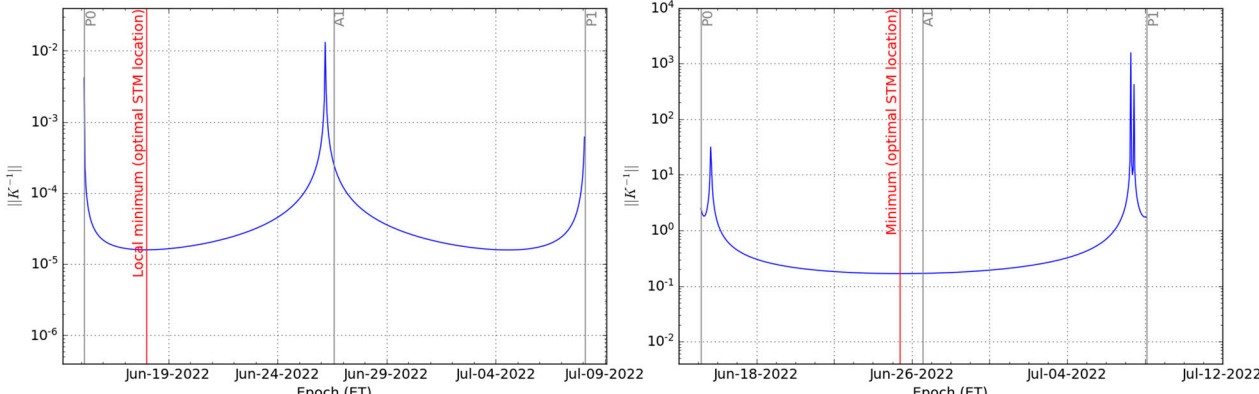

**Figure 7.** K-inverse norm evolution for the targeting of the cartesian position at the apogee A1 (**left**) and the cartesian velocity at the perigee P1 (**right**).

**Table 4.** Statistical ΔV report of the entire mission generated through a Monte-Carlo simulation with 10,000 samples using the optimal trajectory control strategy described in the text.

| Maneuver | Epoch | Aimpoint | Coordinates (EME2000) | ΔV Mean (m/s) | ΔV 99% (m/s) |
|---|---|---|---|---|---|
| OTM1 | $T_D$+22 h | N/A: deterministic open-loop burn | | 11.031 | 11.921 |
| STM1 | OTM1 + 48 h | M0 | B.R, B.T, TCA | 5.706 | 17.306 |
| STM2 | P0−48 h | A1 | X, Y, Z | 4.527 | 18.315 |
| STM3 | P0 + 48 h | A1 | X, Y, Z | 0.405 | 2.195 |
| STM4 | A1 | P1 | VX, VY, VZ | 0.398 | 1.174 |
| STM5 | P1 + 48 h | A2 | X, Y, Z | 0.088 | 0.381 |
| STM6 | A2 | P2 | VX, VY, VZ | 0.106 | 0.312 |
| STM7 | P2 + 48 h | A3 | X, Y, Z | 0.067 | 0.191 |
| STM8 | A3 | P3 | VX, VY, VZ | 0.096 | 0.269 |
| STM9 | P3 + 48 h | A4 | X, Y, Z | 0.061 | 0.153 |
| STM10 | A4 | P4 | VX, VY, VZ | 0.088 | 0.238 |
| STM11 | P4 + 48 h | A5 | X, Y, Z | 0.053 | 0.141 |
| STM12 | A5 | P5 | VX, VY, VZ | 0.086 | 0.245 |
| STM13 | P5 + 48 h | A6 | X, Y, Z | 0.047 | 0.122 |
| STM14 | A6 | P6 | VX, VY, VZ | 0.082 | 0.231 |
| STM15 | P6 + 48 h | A7 | X, Y, Z | 0.073 | 0.216 |
| STM16 | A7 | P7 | VX, VY, VZ | 0.093 | 0.251 |
| STM17 | P7 + 48 h | A8 | X, Y, Z | 0.057 | 0.152 |
| STM18 | A8 | M3 | B.R, B.T, TCA | 0.074 | 0.206 |
| STM19 | P8 + 12 h | M3 | B.R, B.T, TCA | 0.146 | 0.425 |
| Total cumulated statistical ΔV: | | | | 23.287 | 49.443 |

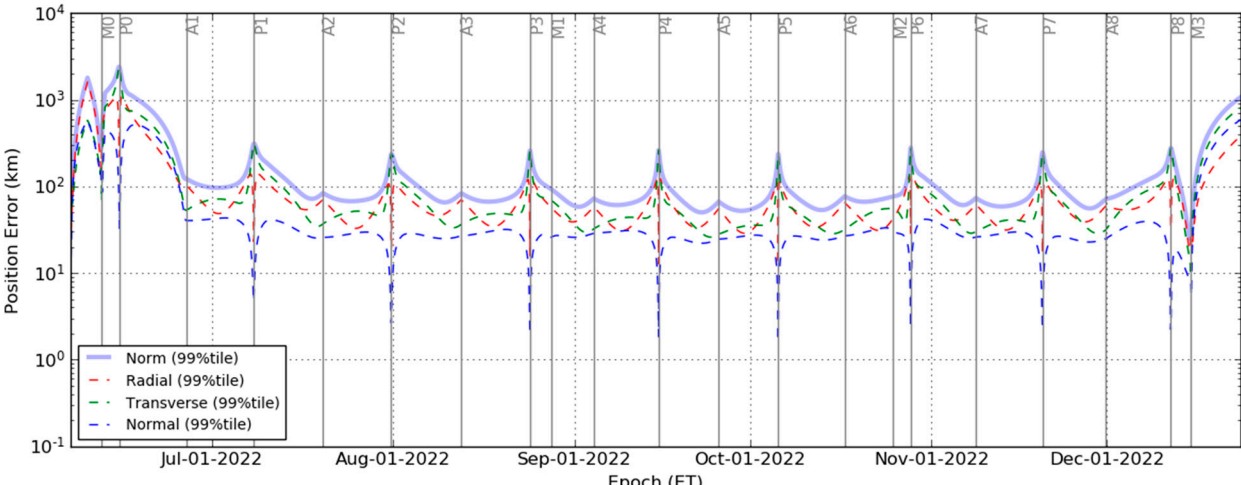

**Figure 8.** Controlled trajectory 99th percentile dispersion through the whole mission using the nominal control strategy. The error is computed with respect to the reference trajectory using the geocentric RTN frame.

As expected, the injection covariance is the primary source of dispersion, and M0 acts as an error amplifier. Unfortunately, as seen in the K-inverse analysis, there is not enough time between the S/C deployment and M0 to add additional correction maneuvers to target the fly-by. Therefore, the ΔV needed to absorb the injection and maneuvers execution errors is distributed through the maneuvers up to the STM4. The rest of the ΔV of the mission is mainly required to correct the maneuvers' execution errors and the OD errors.

## 4. Orbit Determination Analysis

The ArgoMoon OD expected accuracies have been assessed through a covariance analysis. In this context, the OD process is performed using simulated data generated with the same dynamical model adding a synthetic noise. The use of a realistic setup and assumptions allows for inferring the expected accuracy of the estimation. Moreover, by tuning the setup (i.e., *a priori* uncertainties, tracking schedule, observable noise) used to generate and analyze the simulated measurements, it is possible to provide a better understanding of the effects of the main design parameters on the expected OD performances.

### 4.1. Processing Assumptions

The OD is designed as a batch-sequential single-arc estimation where each arc encompasses a trajectory's REV. The S/C state *a priori* covariance at the beginning of the first arc (REV0) is assumed to be equal to the ICPS covariance at the BS1 epoch. The dynamics of the unmodeled proximity flight during the first 30 min of the mission are neglected since it does not significantly increase the state covariance. For REV1 to REV9, the S/C state *a priori* covariance is obtained by mapping the S/C state estimated covariance of the previous arc, scaled by a safety factor of 4. The stochastic accelerations are simulated to account for the mis-modeling of small forces and assess their effects on the expected uncertainties. The measurement of the quality of the estimation will be carried on during the operations by monitoring the estimated stochastic accelerations. The DCO of an orbital maneuver identifies the epoch after which no more data is considered to compute the maneuver. The time between the DCO and the correspondent maneuver shall be sufficient to acquire the necessary data, generate the OD solution, compute the maneuver using the FPC, validate the navigation outputs, generate the S/C commands, and upload them through the DSN antennas, including the necessary margins. Nominally, the DCO of the STMs is set to four days prior to the maneuvers. However, the DCO of STM1 is only 32 h before the maneuver, because it cannot be placed further away from the deployment, and so closer to the Moon flyby, due to ΔV cost reasons. Moreover, since ArgoMoon cannot activate the transponder

during the execution of an orbital maneuver, no tracking data are simulated from five minutes before the maneuver to five minutes after. The introduced baseline assumptions for the OD process are summarized in Table 5.

**Table 5.** Baseline OD assumptions.

| Arc data | Tracking data of a single REV (between two perigees): $P_i \rightarrow P_{i+1}$ | |
|---|---|---|
| Tracking data X/X band | Doppler | 2-way, 60 s of integration time |
| | Range | 2-way, 1 observable every 300 s |
| Data noise and weights | Doppler | 0.1 mm/s at 60 s of integration time (2.81 mHz at X-band) |
| | Range | 2 m |
| Stochastic accelerations | $1.0 \times 10^{-11}$ km/s$^2$ per axis, uncorrelated white noise, 8 h of batch time | |
| Orbital Maneuvers | DCO | 96 h before the maneuver's epoch (nominal) 24 h before the maneuver's epoch (minimum) |
| | Tracking | No tracking data during the maneuver execution |
| REV0 epoch state covariance | ICPS state (Earth-RTN) uncertainty (3-sigma) at BS1 epoch (Table 2) | |
| REV1 to REV9 epoch state covariance | Previous arc's mapped state covariance scaled by a safety factor of 4 | |

### 4.2. Dynamical Model

The dynamical model of the S/C designed for the analysis included the point mass gravitational acceleration due to all the relevant bodies of the Solar System, i.e., the Sun, the planets, and the Moon. The trajectories and the masses of the involved celestial bodies are extracted from the planetary ephemerides NASA/JPL's DE430 [28]. In addition, the extended gravity fields of Earth and Moon are implemented up to degree and order 20, since higher degree terms produce only negligible accelerations on ArgoMoon and drastically increase the computational time. This is because the trajectory is highly elliptical, with perigees altitudes always above 19,000 km. The S/C orbital maneuvers are implemented as impulsive burns since there is no telemetry and communication with the Earth during the execution of the maneuvers. The computation of the Solar Radiation Pressure (SRP) is based on a simplified shape of the S/C, see Figure 9, made of elementary panels with their specific thermo-optical characteristics, as shown in Table 6. The attitude of the S/C is Earth pointing ($-$X body axis points to the Earth) during the simulated tracking passes and Sun-pointing ($-$X body axis points to the Sun) otherwise. The aerodynamic drag due to the Earth's atmosphere is neglected since the altitude at the perigees is expected to be widely outside the exosphere. The primary and secondary albedo, thermal emissions, and thermal recoil pressure, due to the anisotropic emission of thermal radiation are all neglected since they cannot be observed with the expected Doppler noise. The effect of possible mis-modeling is included through solve-for stochastic accelerations, with zero nominal value and an *a priori* uncertainty of $10^{-11}$ km/s$^2$ per axis, constant on 8 h batches, and uncorrelated in time.

### 4.3. Tracking Schedule

The design of the tracking schedule was mainly driven by the DSN availability that changes as a function of the specific mission phases. The tracking availability is up to four hours per day for the first twenty days of the mission (Phase 1 and Phase 2) and then it is reduced to a maximum of four hours per week (Phase 3). In both cases, there is the possibility of distributing the available tracking time into several passes to better cover the trajectory. However, the subdivision was carefully kept to the minimum number of passes sufficient to cover the fundamental events like a fly-by of the Moon, an orbital maneuver, and its DCO epoch.

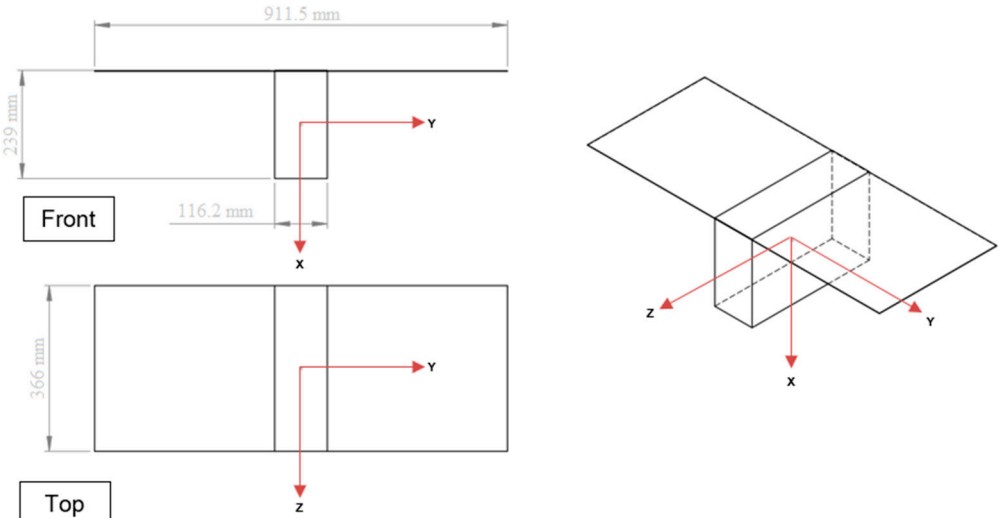

**Figure 9.** ArgoMoon simplified shape model.

**Table 6.** ArgoMoon thermo-optic coefficients.

| Component | Specular Reflectivity ($\rho$) | Diffusive Reflectivity ($\delta$) |
|---|---|---|
| Bus faces | 0.0 | 0.25 |
| Solar arrays | 0.115 | 0.25 |

During the first day after BS1, DSN will allocate one 34-m antenna per complex to track the ICPS and all the deployed CubeSats. The DSN 34-m antennas can provide an uplink to only one S/C at a time, whereas the Multi-Spacecraft Pointing Antenna (MSPA) configuration can provide downlink up to four S/C that lie in the same antenna's beamwidth. The major drawback of the tracking schedule design is that the two-way tracking time, during which radiometric measurements can be collected, must be shared between all CubeSats that will require it. At present, the ArgoMoon tracking schedule foresees 30 min of two-way tracking time per pass during the first day of the mission, with a maximum of two passes per day. Moreover, an additional 1-h tracking pass has been added to cover the OTM1 since it has been found to be critical for maneuver reconstruction.

The Phase 2 tracking schedule consists of two tracking passes per day of two hours each where one is conveniently placed in the morning and the other in the evening (European time). However, the tracking time has been conservatively assumed to be a maximum of one hour per day for the entire REV0 to account for further potential overlapping with other missions as expected by DSN. During Phase 3, the four hours per week of tracking availability have been distributed into two passes per week of two hours each. The two passes were then strategically placed to cover the STMs and their DCO epochs. To account for the sweep and locking time, S/C power budget, and attitude acquisition time, each tracking pass with a duration greater than or equal to two hours was reduced to 1 h and 40 min. The tracking data noise and weights are chosen, considering the expected accuracy obtainable using the DSN tracking and ArgoMoon's IRIS transponder.

*4.4. Filter Configuration*

The OD filter nominal configuration of each arc is reported in Table 7. The filter parameters can be estimated and updated or have just their error considered in the estimated uncertainties. The epoch state *a priori* uncertainty for the REV0 is provided using the full state covariance of the ICPS at the ArgoMoon's deployment epoch, as reported in Table 5. The epoch state covariance of the filter's solution is mapped forward to the next arc epoch and used as *a priori* covariance for the next OD arc, multiplied by a safety factor of 4, which corresponds to a factor 2 in the sigmas. The SRP uncertainty is accounted for by estimating

in each arc a scale factor with a conservative *a priori* σ. The mis-modeling of the SRP and other unmodeled forces is managed by estimating the stochastic accelerations. ArgoMoon's ADCS and PS have not been used before in deep space, then a high conservative *a priori* uncertainty is used in the orbital maneuvers' estimation. The uncertainties related to the masses of the Earth and the Moon from the DE430 ephemeris are considered in the filter. For each tracking pass and station, a stochastic uncorrelated range bias is estimated. The parameters related to the radiometric tracking observables, such as the UT1 time bias, Earth Polar Motion, atmospheric delays, and the DSN station locations, are considered in the filter.

**Table 7.** Baseline filter setup for each OD arc.

| Parameter | | Unit | *A priori* Uncertainty | Estimated/Considered |
|---|---|---|---|---|
| S/C epoch state (REV0) | | - | ICPS state covariance at BS1 (Table 5:) | Estimated |
| S/C epoch state (REV1–REV9) | | - | Estimated covariance mapped from previous arc, multiplied by 4 | Estimated |
| Solar Radiation Pressure Scale Factor | | - | 50% | Estimated |
| Deterministic impulse burns (OTM) | $\Delta V$ | m/s | 10% of nominal | Estimated |
| | Ra | deg | 1.1 | Estimated |
| | Dec | deg | 1.1 | Estimated |
| | Time | s | 3.0 | Estimated |
| Statistical impulse burns (STM) | $\Delta V(X)$ | m/s | 0.011 | Estimated |
| | $\Delta V(Y)$ | m/s | 0.011 | Estimated |
| | $\Delta V(Z)$ | m/s | 0.011 | Estimated |
| | Time | s | 3.0 | Estimated |
| Stochastic accelerations | X/Y/Z | $km/s^2$ | $10^{-11}$, 8-h batches | Estimated |
| Range Bias (per pass) | | m | 2 | Estimated |
| Earth GM | | $km^3/s^2$ | $5.0 \times 10^{-4}$ | Considered |
| Moon GM | | $km^3/s^2$ | $1.4 \times 10^{-4}$ | Considered |
| DSN station locations (per axis) | | cm | 3 | Considered |
| Troposphere path delay (wet/dry) | | cm | 1/1 | Considered |
| Ionosphere path delay (day/night) | | cm | 5/1 | Considered |
| Earth Polar Motion X/Y | | deg | $8.6 \times 10^{-7}$ | Considered |
| UT1 bias | | s | $2.5 \times 10^{-4}$ | Considered |

### 4.5. Baseline Results

The analysis of the OD results is performed by mapping the solution covariance to times of interest and then evaluated in different coordinate sets. In particular, the main quantities to be analyzed are the expected uncertainty in the S/C state in the Radial-Transverse-Normal (RTN) frame centered on the Earth and Moon, the pointing uncertainty from the DSN, and the S/C uncertainty at Moon's flybys in the B-Plane.

The OD analysis has firstly shown that the ArgoMoon pointing uncertainty from the DSN is critical during the first two days of flight. The large *a priori* covariance of the S/C initial state causes the pointing uncertainty to be larger than the 34 m antenna's receiving HPB but lower than the one of the 1.2 m aided acquisition antenna, which will be adopted during the first day of the mission. The optimal delivery schedule has been identified by inspecting the expected pointing uncertainty at each tracking pass using different DCOs. The first delivery to DSN (DEL1) needs to be performed after the OTM1

tracking pass to fulfill the pointing requirement. The DEL1 requires then a DCO after the first tracking pass of the mission to allocate sufficient time for the OD processing. Then, the other deliveries to DSN (DEL2, DEL3, ..., DEL19) are performed before the tracking passes of each STMs using the same DCO allocated to the maneuvers (i.e., DEL2 is performed in correspondence with STM1 with a DCO of 24 h). Finally, the last delivery DEL20 should occur after the reconstruction of the last fly-by M3 to satisfy the requirement up to the EOM. Figures 10 and 11 depict the pointing uncertainty evolution during REV0 and REV1-REV2, respectively.

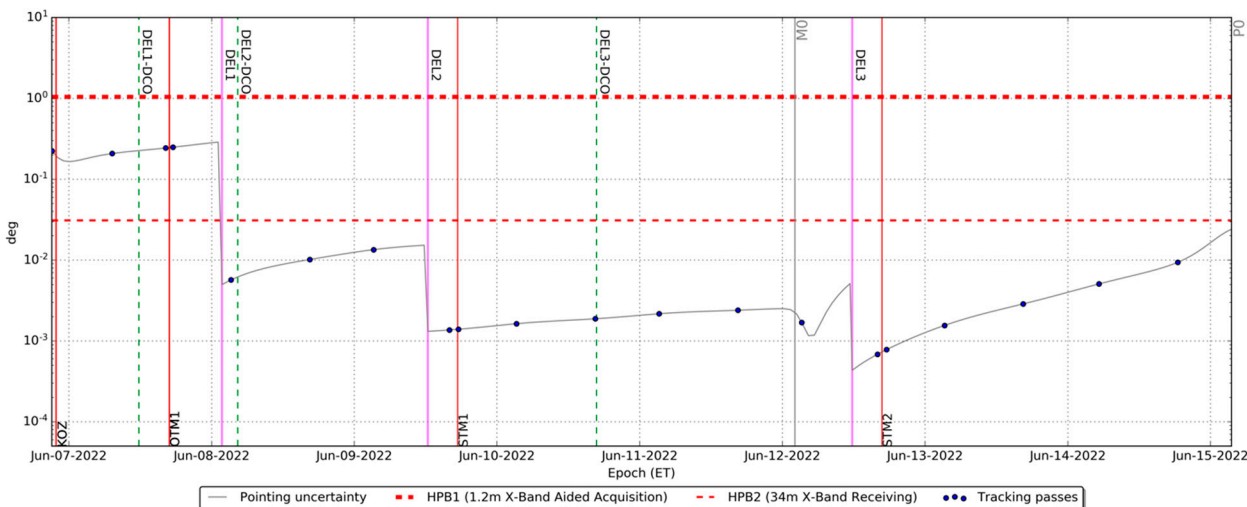

**Figure 10.** DSN to ArgoMoon pointing uncertainty evolution (3-sigma), during REV0. The gray line is the uncertainty evolution using the designed delivery schedule, while the blue dots identify the tracking passes.

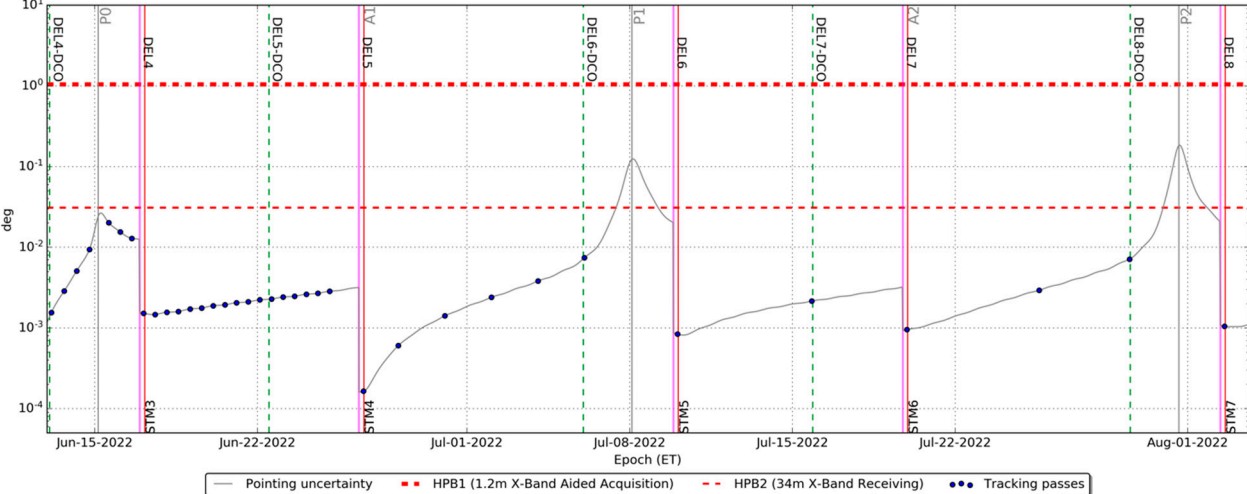

**Figure 11.** DSN to ArgoMoon pointing uncertainty evolution (3-sigma) from P0 to P2 (encompasses REV1 and REV2). The gray line is the uncertainty evolution with the nominal delivery schedule, while the blue dots identify the tracking passes.

The pointing uncertainty through the whole mission exhibits a series of peaks around the perigees that exceed the pointing requirement. This behavior is caused by the rapid decrease of the S/C distance with respect to the Earth during the closest approaches. However, no tracking passes are placed at the perigees and by considering the uncertainty only at the epochs of the tracking passes (the blue dots in Figures 10 and 11), the pointing requirement is always satisfied, at a 3-sigma level.

In Figures 12 and 13, we show the S/C ephemeris uncertainties mapped on the B-Plane of the fly-bys M0 and M3, using different DCOs. The results for M0 highlight that, using the *a priori* uncertainty given by ICPS dispersion, the impact requirement is not satisfied. However, the impact requirement becomes satisfied even by adding just one tracking pass of data after the release. The uncertainty improves by adding more data, and the DCO of the STM1 minimizes the OD error. The expected uncertainties for M3 show that the disposal requirement is always satisfied since the OD error is always contained in the acceptable region at 3-sigma.

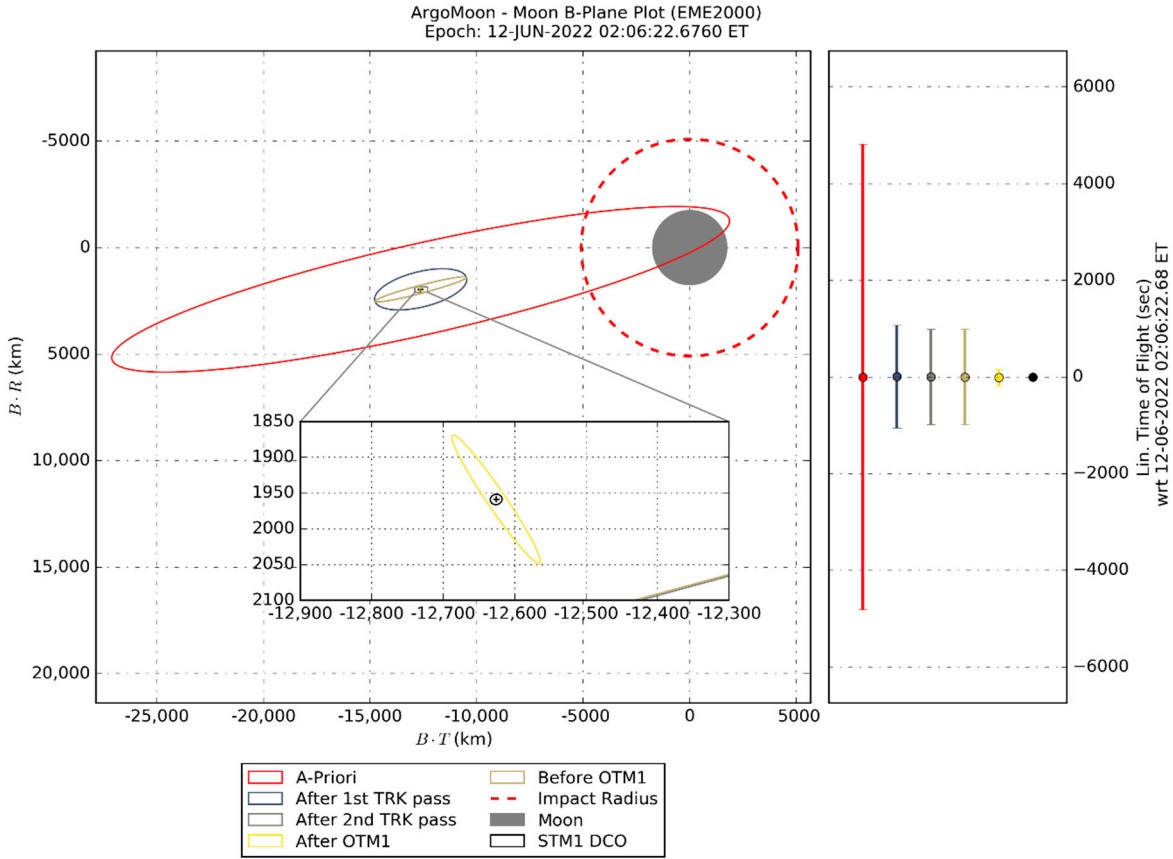

**Figure 12.** B-Plane uncertainties (3-sigma) for the fly-by M0 with DCO at the end of each tracking (TRK) pass up to the DCO of the STM1 (last maneuver before the fly-by). "*A-priori*" refers to the results obtained propagating the ICPS dispersion at the deployment, without processing any data.

Figures 14 and 15 depict the S/C ephemeris uncertainties evolution during REV0 and from P0 to P2 (REV1 and REV2), respectively. The results show rapidly growing uncertainties after the orbital maneuvers because of the large uncertainties adopted in the analysis, partly due to the lack of full performance characterization of the propulsion system. During REV1, whose results are depicted in the first part of Figure 15 (P0 to P1), the ephemeris uncertainties are lower since the tracking schedule is denser than in Phase 2.

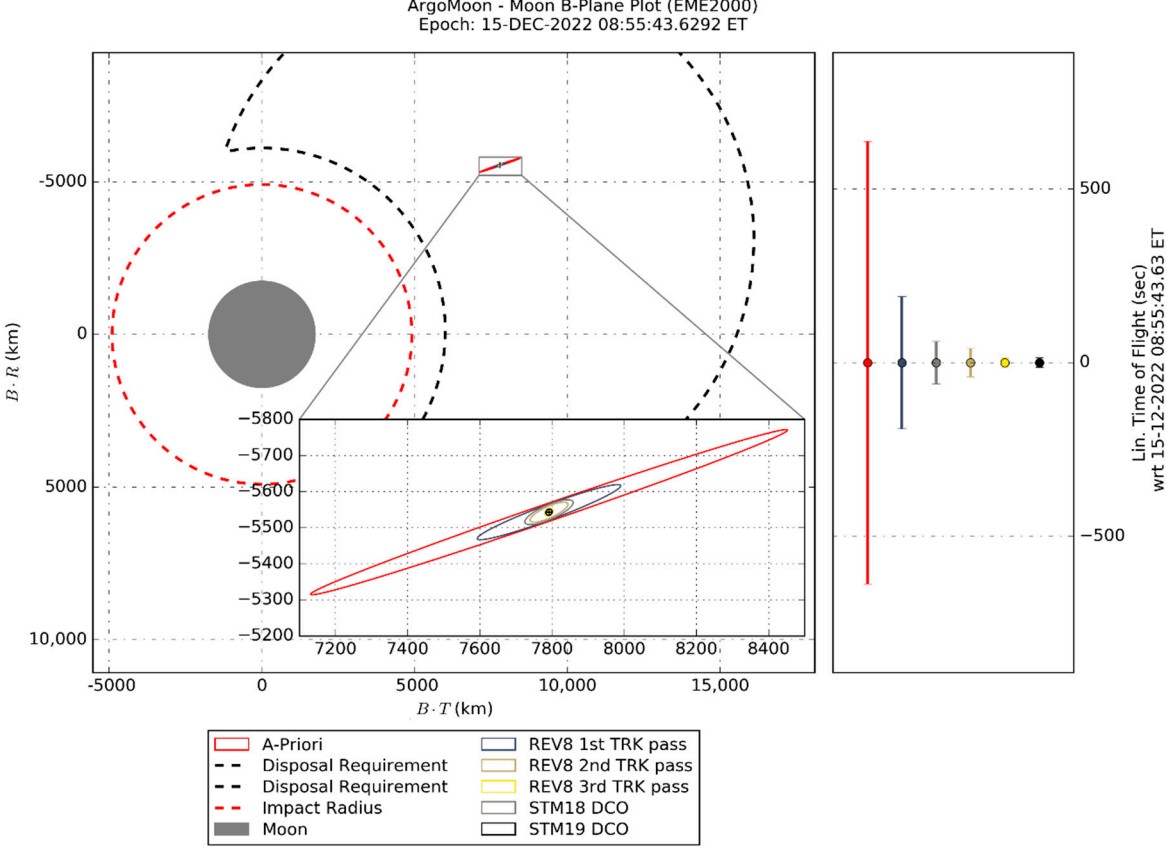

**Figure 13.** B-Plane uncertainties (3-sigma) for the fly-by M3 with DCO at the end of each tracking (TRK) pass up to the DCO of the STM19 (last maneuver before the fly-by). "*A-priori*" refers to the results obtained propagating the expected ephemeris uncertainty at the end of the previous arc (REV7).

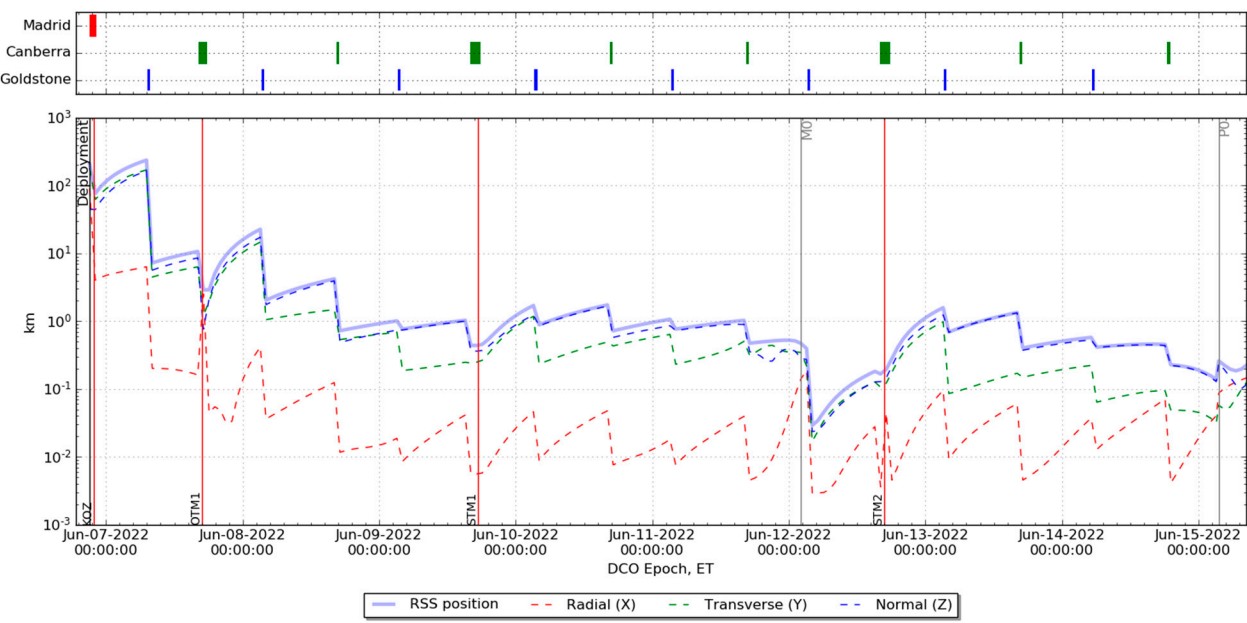

**Figure 14.** Position uncertainty evolution (3-sigma) during REV0, in the Earth-centered RTN frame.

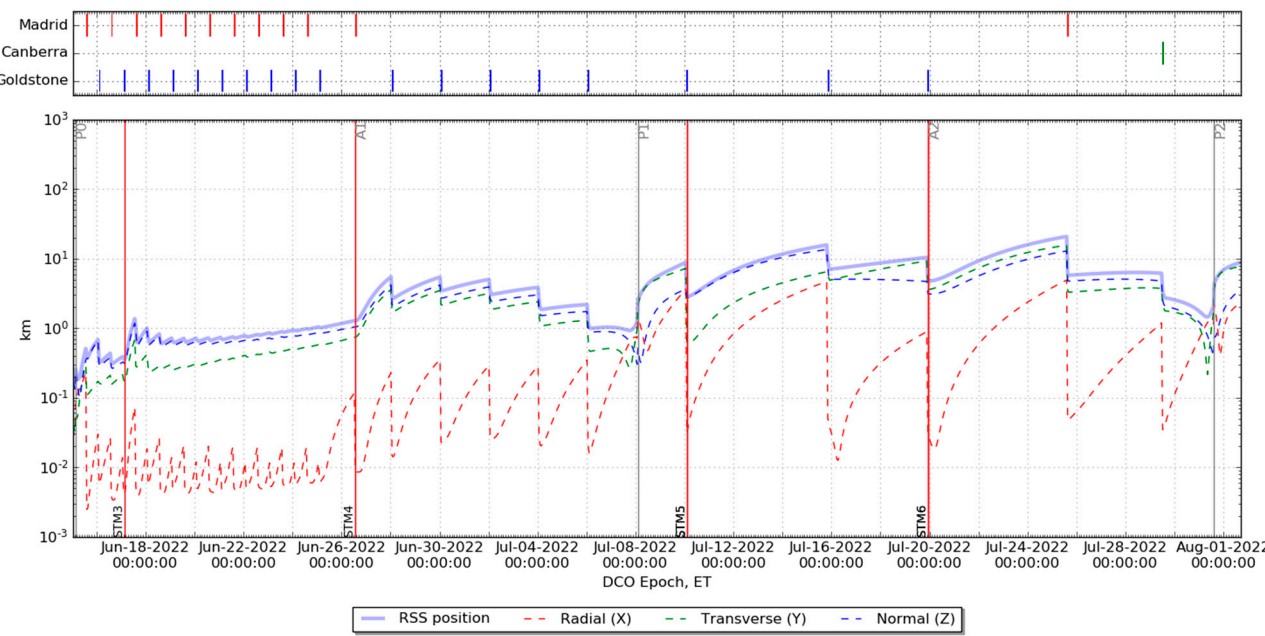

**Figure 15.** Position uncertainty evolution (3-sigma) from P0 to P2, in the Earth-centered RTN frame.

## 5. Sensitivity Analysis

The robustness of the navigation performance has been assessed through a parametric variation of the navigation setup described in Sections 3.2 and 4.4, such as *a priori* uncertainties and measurement acquisition schedule. This sensitivity analysis of the OD and FPC allowed identifying the critical parameters that influence the navigation results, in terms of DSN pointing and disposal requirements (OD) and $\Delta$V cost and trajectory dispersion (FPC). As a result, it has been shown that the dispersion and OD uncertainty at the fly-by M3 are always widely contained in the requirement region (Figure 3) for each of the tested cases, implying that the disposal requirement is not affected by the variation of the parameters. Moreover, even if the trajectory dispersion increases up to a factor of ten in some of the tested sensitivity cases, the impact's risk requirement is always satisfied.

The results of the total statistical $\Delta$V are reported for each relevant case in Table 8. As expected, the injection covariance and the maneuvers execution error are the predominant parameters that drive the total statistical $\Delta$V. In fact, as can be seen in Table 4 and Figure 16, most of the $\Delta$V cost is concentrated in the first four STMs, that are required to absorb the dispersion due to the injection uncertainty (as introduced in Section 3.2).

The targeting of the fly-bys without using the LTOF ("No LTOF targeting" case) allows to save almost 5 m/s at the 99th percentile but at a cost of a slightly larger dispersion at the closest approaches with the Moon. The constraint on the number of tracking passes at the beginning of the mission, the loss of Doppler or Range data, and the enlargement of the OTM1 *a priori* uncertainty barely affect the total statistical $\Delta$V. On the contrary, the former cases do not allow to satisfy the DSN pointing requirement for certain passes during the REV0 and REV1 (in between the passes 0 to 40) as shown in Figure 17. Moreover, the amplification of the *a priori* uncertainties related to the stochastic accelerations and the STM causes an increment of just 3 m/s on the total statistical $\Delta$V but does not permit satisfying the DSN pointing requirement on different tracking passes through the whole mission (i.e., the tracking passes 20, 46, 51, etc.).

**Table 8.** Full mission ΔV statistics for each FPC relevant sensitivity case.

| Case | Mean (m/s) | Sigma (m/s) | ΔV 99% (m/s) |
|---|---|---|---|
| Baseline | 23.3 | 7.6 | 49.5 |
| 0.5 × Injection Covariance | 18.2 | 3.9 | 31.6 |
| 0.5 × Maneuvers Execution Error | 21.4 | 6.2 | 41.3 |
| No LTOF targeting | 22.1 | 7.2 | 46.6 |
| 5 × OTM1 sigmas | 23.3 | 7.7 | 49.5 |
| Maximum 1 pass per day | 23.4 | 7.6 | 49.6 |
| No Doppler data | 23.6 | 7.6 | 49.6 |
| No Range data | 23.5 | 7.6 | 49.8 |
| 10 × Stochastic sigmas | 24.4 | 7.6 | 50.3 |
| 5 × STM sigmas | 26.6 | 7.6 | 52.3 |
| 2 × Maneuvers Execution Error | 30.1 | 12.9 | 77.7 |
| 2 × Injection Covariance | 34.0 | 15.2 | 86.8 |

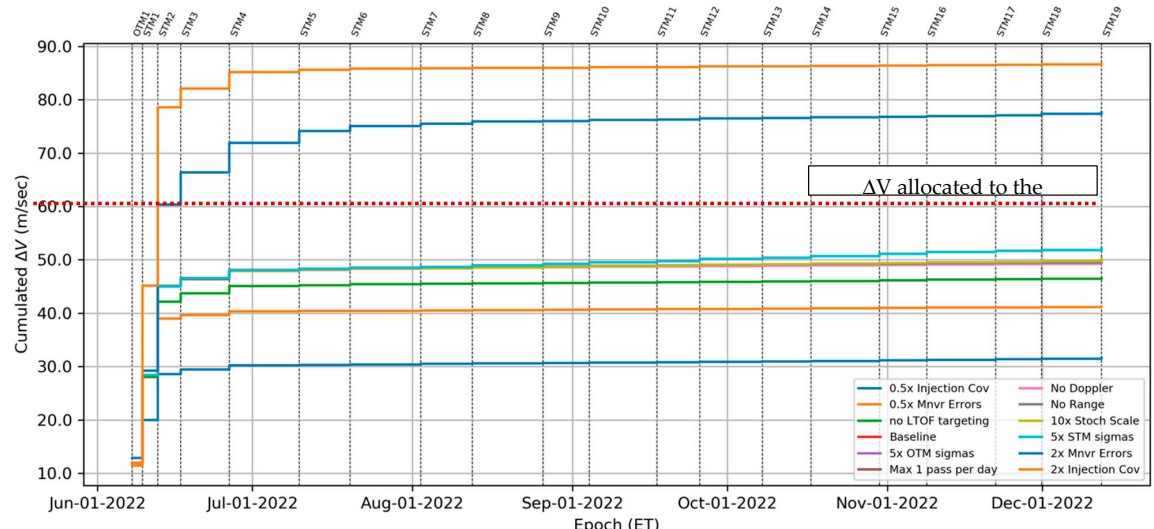

**Figure 16.** Full mission cumulated ΔV statistics plot for each FPC relevant sensitivity case.

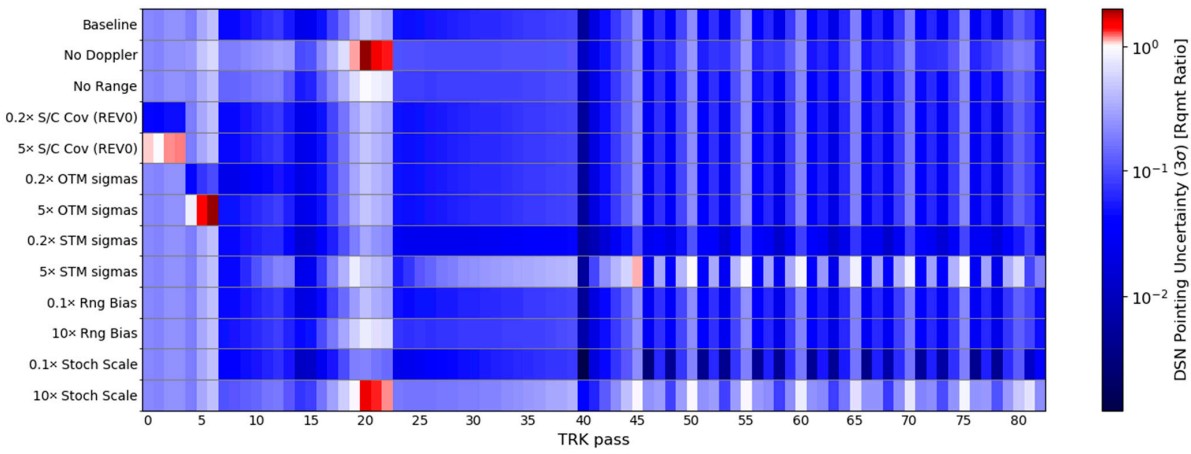

**Figure 17.** Full mission DSN pointing requirement ratio between the expected uncertainty and the requirement, for each tracking (TRK) pass, with respect to each relevant sensitivity case.

Summarizing, the two critical areas for the ArgoMoon navigation are the injection covariance and the maneuvers execution errors. Furthermore, the DSN pointing requirement is also affected by the lack of Doppler data and the increment of mismodeling of the S/C accelerations. The uncertainty on the orbital injection is related to the SLS rocket performance and cannot be controlled, while the performance of the propulsion system wasn't fully characterized due to limited resources available to a CubeSat mission. For these reasons, baseline conservative values have been assumed for the injection covariance, the maneuvers uncertainties, and the stochastic accelerations. Regarding the Doppler observables, since these are important for the reconstruction of orbital maneuvers, a DSN elevated support of Level-3 [29] has been required during the tracking passes that cover the execution of the maneuvers. Thus, the cases that violate the ΔV constraint and DSN pointing requirement are only statistically marginal and have been considered only to understand the effect of the critical parameters on the navigation performance.

## 6. Conclusions

This paper provids an overview of the navigation strategy and expected performance for the ArgoMoon mission. The navigation process has been designed with the aim of reconstructing the S/C trajectory and computing the correction maneuvers to follow the reference trajectory and satisfy the mission requirements. The designed trajectory control strategy allows minimizing the error with respect to the reference trajectory, as well as the statistical ΔV, by using the following targets: the nominal B-Plane coordinates (B.R, B.T, LTOF) for the fly-bys with the Moon, the position at the apogee and the velocity at the perigee when orbiting the Earth. The search of the local minima of the norm of the K-inverse matrix for each target allowed us to find the location of the optimal maneuvers that minimizes the ΔV. The position at the apogee is optimally targeted with a maneuver placed two days after the preceding perigee, while a maneuver at the apogee targets the velocity at the following perigee. From the validated optimal control strategy based on a non-linear Monte Carlo simulation, it was shown that the reference trajectory can be flown with a ΔV of 49.5 m/s at the 99th percentile. Inspection of the OD analysis' results identified the optimal schedule for delivering the estimated trajectory to the DSN required to properly point the antenna toward the S/C. To guarantee a pointing uncertainty from the Earth to the S/C within the beamwidth of the DSN's 34 m antennas, a new estimated trajectory should be provided to DSN at least one time per REV. However, the delivery to DSN during the REV0 should be performed at least three times to satisfy the pointing requirement. Furthermore, due to the large uncertainty occurring in the launch phases, the first two tracking passes have to be supported by the X-band aided acquisition antenna with a larger beamwidth to satisfy the pointing requirement. The S/C ephemeris uncertainty mapped on the B-Plane of the last fly-by shows that the selected trajectory and navigation strategy can fulfill the disposal requirement with a 99% confidence. The sensitivity analysis pointed out that the trajectory controllability is strongly dependent on the injection error and the performance of the orbital propulsion system. The DSN pointing requirement is instead affected by the uncertainties on the stochastic accelerations, orbital maneuvers, and if there is a shortage in tracking data. However, the worsening in the S/C pointing uncertainty is a behavior that can be controlled by varying the schedule of the deliveries to DSN. Finally, all the tested cases of the sensitivity analysis have widely satisfied the impact's risk and disposal requirements. The pictures of the Earth and the Moon that will be acquired by ArgoMoon's cameras may be processed on the ground and used as optical observables (centroids) to further increase the robustness of the navigation. Summarizing, the analysis described in this paper demonstrated that the navigation of ArgoMoon is feasible, under realistic assumptions on the mission scenario and the technological capabilities of the space and ground segment. The main challenges are related to the performance of the propulsion system as well as the injection accuracy provided by SLS.

**Author Contributions:** Conceptualization, M.L., M.Z. and B.C.; Formal analysis, M.L.; Investigation, M.Z., B.C., S.S. and S.P. (Silvio Patruno); Methodology, M.L.; Project administration, M.Z., P.T., V.D.T., B.C. and S.P. (Simone Pirrotta); Resources, M.L., S.S. and S.P. (Silvio Patruno); Software, M.L., I.G., L.G.C., E.G. and R.L.M.; Supervision, M.Z., P.T., V.D.T., S.P. (Silvio Patruno) and S.P. (Simone Pirrotta); Validation, M.Z., I.G., L.G.C. and E.G.; Writing—original draft, M.L.; Writing—review & editing, M.Z., P.T., B.C. and S.S. All authors have read and agreed to the published version of the manuscript.

**Funding:** This research was funded by Agenzia Spaziale Italiana, grant number 2019-31-HH.0 CUP F84I190012600 and by Argotec, grant number ARG-CON-AGM-0319.

**Data Availability Statement:** No new data were analyzed in this study. We only simulated tracking data from the ArgoMoon CubeSat, thus data sharing is not applicable to this article.

**Acknowledgments:** I.G., L.G.C., E.G., R.L.M., M.L., P.T. and M.Z. wish to acknowledge Caltech and the Jet Propulsion Laboratory for granting the University of Bologna a license to an executable version of MONTE Project Edition S/W.

**Conflicts of Interest:** The authors declare no conflict of interest.

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
