# Peer review of "Design and Analysis of the Cis-Lunar Navigation for the ArgoMoon CubeSat Mission"

_aerospace, doi:10.3390/aerospace9110659_

Round 1
Reviewer 1 Report
Dear authors,
this is a very interesting and sound analysis of the selected topic. Thanks for providing it a already very high developed quality level. My background is not radio science but I know about OD and tracking with other techniques and I can follow all points, methods, approaches and results almost completely. It was nice to read and I learned a lot.
I have quite some comments but I would rate them as quite minor as they are only questions or small points about understanding specific things. I expect all the issues can be solved using just minor changes or clarifications. The conclusion had a few sentences that read a bit weird but with minor changes, e.g. as suggested, this can be easily improved.
This is my list:
L. 62: How is navigation/orientation/positioning established during the first 30 mins of the mission, during the ProxOps phase?
L. 65: what does “deterministic orbital maneuver” mean?
L. 92: What is the FOV of the two cameras?
L. 99: “can be considered for past and future …”
L. 102: “it is trimmed” … how does the SC trim the orbit to shape the orbit around the Earth? Do you mean the SC trims the orbit during the flyby by using propulsion to finalize the Earth orbit insertion? This is only a little bit unclear.
Figure 2, upper image: the symbols used to mark the beginning and the end of the trajectory are not well choosen. One can barely see the green begin but cannot find the red end symbol. Also the different colors for SC and Moon orbit are difficult to distinguish. Please improve the selection a bit.
Table 3: Could you use relative times for all relevant dates (e.g. BS1 + 120 days) because the absolute ones are already not valid anymore?
L. 110: what is the monte carlo simulation exactly used for or why is it necessary to do that? Maybe add a ref that this will be explained with more detail later in section XY if applicable?
L. 112: what about the maneuvering during the 30 min (ProxOps) and the related uncertainties? How does the autonomous operation work or on what is it based? Maybe just add a ref that this phase will be described in sec XY if this is applicable here?
L. 142: a few background facts on the done Monte Carlo simulation or a reference to where it is described in the paper would be good.
L. 168: Are the ranges integrated over a certain timespan as well? Are they two-way, yes right? What are a few meters – up to 5 or up to 10? Is this accuracy or precision?
L. 173: What are the approximate expected uncertainties of the main error sources?
L. 174: Are the orbital maneuvers or the potential error sources purely statistical?
L. 175: “computed a priori. Then, …” Then -> Thus
L. 178: The STM are scheduled a priori? But it is not executed necessarily? Earlier you said the STMs cannot be calculated a priori. I think the formulation is just a little bit unclear here.
L. 237: “after M0 is and indicator” and -> an?
L 243: “This required to find” -> “this requires to find”?
L. 256: Of course you already introduced the names of the most relevant points in a table earlier, but you could add them in brackets in this sentence again, so a reader knows more quickly which one it is when looking at the figures.
L. 271: The text says P7 but the image starts with A8 and only P8 is shown. Something missing or wrong here?
L. 273: I don’t fully get the summary. At first, you use this analysis to optimize the points in time/positions to put the STM’s. This could be said somewhere earlier in particular (e.g. adding a sentence in line 243) to easier get what you do with this analysis – if I get it right. Further, you found the best positions to be 48h after OTM1, after A8 and after P8. This is not exactly the apogee and perigee to my understanding, which is why I got a bit confused by your summary sentence. Maybe the timeframes are so short that it still considered apo- or perigee?
Fig. 6 and 7: So Figure 6 shows the K^-1 for two positions and Fig. 7 for position (left) and velocity (right)?
L. 277: So for Figure 7 the message is the two optimum points are “inverse” either at the Jun 18th or 26th? Is that correct? So for which did you decide in the end and why?
Fig. 8: To my feeling it would make sense to also add the STM’s to this figure because you discuss them up to STM4 (L. 294). What do you think?
Table 5: Would it make sense to add one comment about the difference in delta V between the mean and the 99% solution, also w.r.t. the total available delta V of the SC? Or is this what you mean with the comment “The rest of the 294 statistical ΔV is mainly required to correct the maneuver execution errors and the OD errors.”?
Table 6: Last line says a safety factor of 2 but the text says a factor of 4 in line 313!? Also, would it make sense to add the assumed radio observables uncertainties here again?
L. 331: Please add the reason why the higher order grav. Field coeff do not cause relevant errors.
L. 335: Is the SC orientation always the same during the mission or are changes represented in the estimation?
Table 9 and Fig. 16: So only two of the listed cases have a delta V larger than available. What does this mean? Are they likely to happen? Also I feel like a summary statement in Line 466 could be missing w.r.t. all the selected and calculated cases. They are carefully set up, calculated and analyzed but what does the result mean w.r.t. mission? Is the big statement that almost all cases fulfill the available delta v limit, even though it has to be taken care for some special cases cause they might violate the DSN pointing accuracy? Something final like this might be good here, even if it is just one sentence.
L. 475: “if the following targets are used” -> “ if the errors at the following points are minimized”? Otherwise the sentence sounds like something is missing “if the following targets are used: the nominal coordinates” … I would assume that this would be the case anyways?
L. 478: “the optimal maneuvers location that minimizes the” -> “the optimum maneuver locations that minimize the …”?
L. 478: “The position at the apogee is optimally targeted two days after the preceding perigee, while the targeting of the velocity at the perigee is optimal if the maneuver is located at the preceding apogee.” -> “The position at the apogee is optimally targeted two days after the preceding perigee with a maneuver, while the targeting of the velocity at the perigee is optimal at the preceding apogee.”
L. 480: “The validation of the optimal control strategy performed using a non-linear Monte Carlo simulation showed that the reference trajectory can be flown” -> “From the validated optimal control strategy based on a non-linear Monte Carlo simulation, it was shown that the reference trajectory can be flown”?
L. 486: “Furthermore, due to the large uncertainty occurring in the launch phases, the first two tracking passes have to be supported by the X-Band aided acquisition antenna [12] with a larger beamwidth without which the pointing requirement can’t be satisfied” -> “Furthermore, due to the large uncertainty during the launch phases, the first two tracking passes have to be supported by the X-Band aided acquisition antenna [12] with a larger beamwidth to satisfy the pointing requirement”
L. 489: “the OD can satisfy”? Would it not be the selected trajectory or mission profile?
L. 492: “uncertainties on the stochastic accelerations, the orbital maneuvers, and the lack of tracking data” -> “uncertainties on the stochastic accelerations, orbital maneuvers and if there is a shortage in tracking data”?
Reviewer 2 Report
Please see the attached document.

Reviewer 3 Report
The ArgoMoon mission will be the first European CubeSat that will fly in deep space. This paper presents an overview of the navigation design and analysis.
1. The introduction seems a little short. I suggest adding description about new features in navigation of the ArgoMoon CubeSat, or the difference in design the navigation between the ArgoMoon CubeSat and previous spacecraft.
some papers about missions to the moon are also suggested to be cited, e.g.
[1] De Grossi, F., Marzioli, P., Cho, M., Santoni, F., & Circi, C. (2021). Trajectory optimization for the Horyu-VI international lunar mission. Astrodynamics, 5(3), 263-278.
[2] Wang, Z. S., Meng, Z., Gao, S., & Peng, J. (2021). Orbit Design Elements of Chang’e 5 Mission. Space: Science & Technology, 2021.
2. Please give more details about how to search and determine the optimal aimpoints.
3. What method or solver do you use to propagate the trajectories?
Round 2
Reviewer 1 Report
No more comments ... is good to go!
Reviewer 2 Report
Great paper!
Reviewer 3 Report
My comments have been addressed.